# Quercetin treatment reduces the severity of renal dysplasia in a beta-catenin dependent manner

Joanna Cunanan, Erin Deacon, Kristina Cunanan, Zifan Yang, Antje Ask, Lily Morikawa, Ekaterina Todorova, Darren Bridgewater *

Department of Pathology and Molecular Medicine, McMaster University, Hamilton, Canada

* bridgew@mcmaster.ca

**Data Availability Statement:** All data are within the paper and its Supporting Information files.

## Abstract

Renal dysplasia, the major cause of childhood renal failure, is characterized by defective branching morphogenesis and nephrogenesis. Beta-catenin, a transcription factor and cell adhesion molecule, is markedly increased in the nucleus of kidney cells in human renal dysplasia and contributes to its pathogenesis by altering target genes that are essential for kidney development. Quercetin, a naturally occurring flavonoid, reduces nuclear beta-catenin levels and reduces beta-catenin transcriptional activity. In this study, we utilized wild type and dysplastic mouse kidney organ explants to determine if quercetin reduces beta-catenin activity during kidney development and whether it improves the severity of renal dysplasia. In wild type kidney explants, quercetin treatment resulted in abnormal branching morphogenesis and nephrogenesis in a dose dependent manner. In wild type embryonic kidneys, quercetin reduced nuclear beta-catenin expression and decreased expression of beta-catenin target genes *Pax2*, *Six2*, and *Gdnf*, which are essential for kidney development. Our $RD^B$ mouse model of renal dysplasia recapitulates the overexpression of beta-catenin and histopathological changes observed in human renal dysplasia. $RD^B$ kidneys treated with quercetin resulted in improvements in the overall histopathology, tissue organization, ureteric branching morphogenesis, and nephrogenesis. Quercetin treatment also resulted in reduced nuclear beta-catenin and reduced *Pax2* expression. These improvements were associated with the proper organization of vimentin, NCAM, and E-cadherin, and a 45% increase in the number of developing and maturing nephrons. Further, our results show that in human renal dysplasia, beta-catenin, vimentin, and e-cadherin also have abnormal expression patterns. Taken together, these data demonstrate that quercetin treatment reduces nuclear beta-catenin and this is associated with improved epithelial organization of developing nephrons, resulting in increased developing nephrons and a partial rescue of renal dysplasia.

**Funding:** DB-Canadian Institutes of Health Research; DB-Kidney Foundation of Canada; DB-National Science and Engineering Research Council.

**Competing interests:** No authors have competing interests.

## Introduction

Renal dysplasia is a developmental disorder of the kidney and affects approximately 0.1% of live births and 2% at paediatric autopsy [1–5]. Renal dysplasia accounts for 30–40% of end stage renal disease in children and also contributes to adult onset diseases such as chronic renal insufficiency, hypertension, and stroke, especially in individuals under the age of 25 [6–8]. Renal dysplasia encompasses a broad range of gross and histopathological abnormalities [1–5]. At the gross level, there can be a complete absence of kidney tissue (renal agenesis), abnormally small kidneys (renal hypoplasia), abnormally large kidneys (renal hyperplasia), multiple kidneys fused together (multiplex kidneys with multiple ureters), and abnormally large kidneys with cystic transformation (multicystic dysplasia). At the histological level, dysplastic kidneys can exhibit disorganized and incomplete collecting duct and nephron formation, poorly differentiated epithelial tubules surrounded by a fibromuscular collar, metaplastic cartilage transformation, cystic glomeruli, and expanded loosely packed renal stroma. These abnormalities can be unilateral or bilateral (affecting one or both kidneys) and can be diffuse (involving the entire kidney), segmental (involving segments of the kidney) or focal (affected regions are surrounded by normal tissue) [1–5].

The broad range of macroscopic and histopathological phenotypes observed during renal dysplasia result from abnormalities in kidney development [8]. Normal kidney development occurs through the interactions of the ureteric epithelium, metanephric mesenchyme, and renal stroma [9–11]. The interactions between these cells result in branching morphogenesis and nephrogenesis. At embryonic day (E) 10.5 in mice or 6–8 weeks in humans, an outgrowth of ureteric epithelial cells buds off of the caudal region of the Wolffian duct. In response to signals from the neighbouring metanephric mesenchyme, the ureteric epithelial cells elongate and migrate into the adjacent pool of metanephric mesenchyme cells. Once in the mesenchyme, the ureteric epithelium tips proliferate, expand, and elongate to form branches. This bifid branching pattern occurs for 10 branch generations in mice and 15 branch generations in humans to form 15,000 or 60,000 collecting ducts in mice and humans, respectively. While undergoing branching morphogenesis, the ureteric epithelium sends signals to the metanephric mesenchyme to undergo nephrogenesis, the formation of the nephrons. The mesenchymal cells cluster and organize along the ureteric epithelium tips, undergo mesenchymal-to-epithelial transition, and progress through several distinct morphological stages to form approximately 10,000 nephrons in mice and 1 million nephrons in humans [9–11].

Beta-catenin is a multifunctional protein found in the cell membrane, cytoplasm, and nucleus. The membrane-bound pool of beta-catenin links E-cadherin to the actin cytoskeleton and facilitates epithelial adhesion and epithelial morphogenesis. In the cytoplasm, beta-catenin is a key signaling molecule that transmits external signals to the nucleus for various signaling pathways. In the nucleus, beta-catenin is a co-transcriptional activator that binds to several co-activators (i.e. Tcf/Lef) to regulate gene expression. An imbalance of the beta-catenin intracellular pools is associated with various disease states, including abnormal organogenesis [12, 13]. Our laboratory has demonstrated that beta-catenin is overexpressed in human renal dysplasia. Specifically, the overexpression is observed primarily in the nucleus of the metanephric mesenchyme, ureteric epithelium, and renal stroma cells [14–16]. The generation of transgenic mouse models with nuclear and cytoplasmic beta-catenin overexpression in the mesenchyme, epithelium, or renal stroma of the developing kidney exhibit gross and histopathological changes indistinguishable to that observed in human renal dysplasia [14–16]. These abnormalities result primarily from nuclear beta-catenin disrupting the expression of genes that are essential for kidney development (i.e. *Gdnf*, *Gata3*, *Pax2*) [14–16].

Our previous studies suggest that reducing nuclear beta-catenin levels or reducing beta-catenin transcriptional activity in dysplastic kidneys could be a therapeutic target for renal dysplasia. In support of this, there are several recent studies that found natural compounds effectively reduce beta-catenin signaling in kidney diseases such as renal fibrosis and chronic kidney disease. These natural products include the plant-derived compounds 25-*O*-Methylalisol F, Alisol B 23-acetate, porioic acid, ergone, and quercetin [17–20]. Quercetin is a flavonoid found in various fruits and vegetables and is one of the most abundant natural plant polyphenols found in the human diet [21]. Quercetin has been widely researched for its diverse therapeutic properties, such as anti-inflammatory [22–24], antioxidant [24, 25], anti-microbial [26], anti-hypotensive and anti-atherogenic [27], as well as protecting against metabolic and aging-related diseases [24, 28] and protecting cells against high glucose induced damage [29]. Further, studies have also demonstrated that quercetin inhibits nuclear beta-catenin accumulation and decreases the transcriptional activity of beta-catenin target genes in colon, pancreatic, and breast cancer cells [21, 30, 31]. For example, quercetin treatment of breast cancer cells that are associated with increased nuclear beta-catenin and transcriptional activity resulted in decreased nuclear beta-catenin and decreased expression levels of *cyclin D1* and *c-myc*. This resulted in dose-dependent decreases in cell survival, migration, and invasiveness of the cancer cells [31]. Similar results were observed in pancreatic and colon cancer cells [21, 30]. Further, in a mouse model of acute kidney injury resulting from unilateral ureter obstruction, intraperitoneal injections of mice with quercetin resulted in decreased levels of nuclear beta-catenin compared to vehicle treated controls [20]. This resulted in a downregulated expression of markers for fibroblast activity (including fibronectin and α-smooth muscle actin), and attenuated the progression of renal interstitial fibrosis [20]. Taken together, these studies demonstrate that quercetin reduces nuclear beta-catenin and transcriptional activity resulting in improved outcomes in disease models and is therefore an ideal candidate to determine its therapeutic effects on renal dysplasia caused by abnormal nuclear beta-catenin levels and activity.

In the present study, we hypothesize that quercetin can improve the pathological changes observed in renal dysplasia by modulating the expression of beta-catenin. Our results demonstrate that quercetin treatment of kidney organ explants reduces nuclear beta-catenin expression and the expression of genes essential for kidney development. Our results also demonstrate that quercetin treatment reduces the severity of renal dysplasia in kidney explants from our mouse model of severe renal dysplasia. We demonstrate that quercetin functions by redistributing beta-catenin and E-cadherin to the cell membrane, which is required for both developing nephron structures and mesenchyme-to-epithelial transition. These changes were associated with a 45% increase in the number of normally developing nephrons. These results demonstrate for the first time that quercetin affects normal kidney development and rescues renal dysplasia by modulating beta-catenin expression and gene transcription.

## Materials and methods

### Mouse strains

The mouse model of renal dysplasia was generated by crossing male *RarB2Cre* mice (metanephric mesenchyme specific Cre expression) [14] with female mice containing *LoxP* sites flanking exon 3 of the beta-catenin allele [32]. This cross excises phosphorylation sites in beta-catenin that prevent its degradation. The resulting cross generates mutant embryos with beta-catenin accumulating in the cytoplasm and nucleus of the metanephric mesenchyme (termed $RD^B$), as previously described [14]. Twelve PM on the day of vaginal plug detection is embryonic day (E) 0.5 and all pregnant dams were euthanized via cervical dislocation at E13.5. A total of 26 wild type CD1 mice and 35 $RD^B$ adult mice were used for this study. CD1 wild type

mice were purchased from Charles River Laboratories (Wilmington, MA, USA), *RarB2Cre* male mice were gifted by Dr. Norman Rosenblum from The Hospital for Sick Children (Toronto, ON, Canada), and *LoxP* female mice were purchased from Jackson Laboratories (Bar Harbor, ME, USA). All animal studies were performed in strict accordance with institutional animal care guidelines as set by the Hamilton Health Sciences Integrated Research Ethics Board and Animal Research Ethics Board at McMaster University (Animal Utilization Protocol #18-03-12).

## Embryonic kidney organ culture

Wild type and $RD^B$ mutants were micro-dissected from E13.5 embryos and placed on 3µm polyethylene terephthalate transwell filters (Sigma Aldrich, Oakville, ON, Canada) in 6-well culture plates. Media contained Dulbecco's minimal essential medium (ThermoFisher Scientific, Mississauga, ON, Canada) with 1% penicillin/streptomycin (ThermoFisher Scientific, Mississauga, ON, Canada) and supplemented with or without quercetin, catalog # Q4951 (Sigma Aldrich, Oakville, ON, Canada) for 48 hours in 37˚C and 5% $CO_2$. Quercetin was initially prepared as a 50mM stock solution in 100% dimethyl sulfoxide (Sigma Aldrich, Oakville, ON, Canada).

## Human kidney dysplastic tissue

Archived paraffin-embedded human dysplastic kidney tissues were obtained from the McMaster University (Hamilton, ON, Canada) pathology department and used for research purposes in strict accordance with guidelines approved by the Hamilton Health Sciences Integrated Research Ethics Board (approval #13-160-T).

## Tissue preparation, histology, and immunostaining

Kidneys were fixed in 4% paraformaldehyde at 4˚C for 48 hours and then paraffin-embedded. Kidneys were sectioned to 5µm on a microtome and mounted on Superfrost microscope slides (VWR, Mississauga, ON, Canada). Tissue sections were dried overnight at room temperature and then H&E stained (Sigma Aldrich, Oakville, ON, Canada). For immuno-staining, slides were deparaffinized and re-hydrated using xylene and graded ethanol washes (100%, 95%, 70% and 50%), washed in phosphate-buffered saline (PBS) and antigen retrieval was performed for 5 minutes in 11.4 mmol/L sodium citrate buffer solution (pH 6.0) in a pressure cooker. For immunofluorescence, samples were blocked with 7.5% normal goat serum (Sigma Aldrich, Oakville, ON, Canada) and 4.5% bovine serum albumin (GE Healthcare, Mississauga, ON, Canada) at room temperature for 1 hour, followed by incubation with primary antibodies cytokeratin (Sigma Aldrich, Saint Louis, MO, USA, 1:200 dilution), Pax2 (Biolegend, San Diego, CA, USA, 1:200 dilution), Six2 (Abcam, Cambridge, MA, USA, 1:200 dilution) at 4˚C overnight. Primary antibodies were diluted in an incubation buffer containing 1% PBS, 3% bovine serum albumin, 5% normal goat serum and 0.3% Tween20. Tissue samples were washed 3 times in PBS and incubated with secondary antibodies Alexa Fluor 488 anti-rabbit and DyLight 594 anti-mouse (Invitrogen, Carlsbad, CA, USA, 1:1000 dilution) in incubation buffer at room temperature for 1 hour, washed in PBS, counterstained with DAPI (Sigma Aldrich, Oakville, ON, Canada), washed and fixed using Fluoromount mounting medium (Sigma Aldrich, Oakville, ON, Canada). Immunohistochemistry was performed using the Vectastain Elite avidin-biotin complex kit (Vector Laboratories, Burlingame, CA, USA). After antigen retrieval, slides were immersed in 3% hydrogen peroxide at room temperature for 30 minutes followed by 5% normal goat serum (Sigma Aldrich, Oakville, ON, Canada) in phosphate-buffered saline with Tween20 (PBST) at room temperature for 30 minutes. Avidin and

biotin blocking reagents (Vector Laboratories) were then applied as per manufacturer protocols. For mouse monoclonal primary antibodies, mouse-on-mouse blocking reagents (Vector Laboratories) were used as per manufacturer protocols. Primary antibodies used were beta-catenin (BD Biosciences, San Jose, CA, USA, 1:200 dilution), vimentin (Santa Cruz Biotechnology, Dallas, TX, USA, 1:50 dilution), NCAM (Sigma Aldrich, Saint Louis, MO, USA, 1:100 dilution) and E-cadherin (Abcam, Cambridge, MA, USA, 1:100 dilution). Primary antibodies were incubated overnight at 4˚C. Samples were washed with PBS then incubated with biotinylated anti-mouse secondary antibody (Vector Laboratories, Burlingame, CA, USA, 1:200 dilution) in PBST at room temperature for 45 minutes. Avidin-Biotin Complex (ABC) reagent (Vector Laboratories) was applied on the samples followed by colorimetric visualization using diaminobenzidine (DAB; Vector Laboratories) as per manufacturer protocols. Slides were dehydrated using graded ethanol washes (50%, 75%, 95%, and 100%) and xylene washes, followed by mounting using Vectamount mounting medium (Vector Laboratories). All images were acquired using the Olympus BX80 light and fluorescence microscope and CellSens image acquisition software (Olympus Life Science, Waltham, MA, USA).

## Whole-mount immunofluorescence

Cultured kidney explants were fixed in 100% methanol at -20˚C for 24 hours. Kidneys were washed in sterile PBS (pH 7.4) and blocked in 10% normal goat serum (Sigma Aldrich, Oakville, ON, Canada) for 30 minutes. Cytokeratin (Sigma Aldrich, Saint Louis, MO, USA, 1:200 dilution) and Pax2 (Biolegend, San Diego, CA, USA, 1:400 dilution) primary antibodies diluted in an incubation buffer containing 1% PBS, 3% bovine serum albumin, 5% normal goat serum and 0.3% Tween20 were applied and incubated at 37˚C for 1.5 hours. Kidneys were washed 3 times in sterile PBS, then incubated with secondary antibodies Alexa Fluor 488 anti-rabbit and DyLight 594 anti-mouse (Invitrogen, Carlsbad, CA, USA, 1:300 dilution) in incubation buffer at 37˚C for 1.5 hours. Branching morphogenesis was imaged on Olympus fluorescence microscope (Olympus Life Science, Waltham, MA, USA).

## Western blot

A minimum of 10 kidneys for each treatment group were snap-frozen and homogenized in radioimmunoprecipitation assay (RIPA) lysis buffer (Sigma Aldrich, Oakville, ON, Canada) with 1% protease inhibitor cocktail and 1% phosphatase inhibitor cocktail (ThermoFisher Scientific, Mississauga, ON, Canada). Samples were centrifuged at 13,000rcf and supernatants were collected. Protein concentration was determined by Bradford assay (BioRad Laboratories, Mississauga, ON, Canada). Western blot was performed using 13μg of total protein added to 4X Laemmli sample loading buffer and heat denatured at 100˚C for 5 minutes. SDS-PAGE was performed using 10% Mini-Protean pre-cast gels (BioRad Laboratories, Mississauga, ON, Canada) followed by electrotransfer onto nitrocellulose blotting membrane (GE Healthcare, Mississauga, ON, Canada). Blots were blocked in 5% skimmed milk in tris-buffered saline with Tween20 (TBST) followed by incubation with beta-catenin primary antibody (BD Biosciences, San Jose, CA, USA, 1:2000 dilution) diluted in 5% bovine serum albumin in TBST, overnight at 4˚C. After washing in TBST, blots were incubated in secondary HRP-conjugated anti-mouse antibody (Abcam, Cambridge, MA, USA, 1:2000 dilution) for 1 hour at room temperature. The reaction was visualized using Pierce ECL Western Blotting Substrate (ThermoFisher Scientific, Mississauga, ON, Canada) as per manufacturer instructions and imaged using GeneSys chemiluminescence imaging system (Syngene, Frederick, MD, USA). SDS-PAGE gels were stained with Coomassie Brilliant Blue R-250 (ThermoFisher Scientific, Mississauga, ON, Canada) to visualize protein loading and Western blots were normalized using total

protein as the denominator, as per recommended protocols [33]. Quantification was performed using densitometry analysis on ImageJ (version 1.52; Bethesda, MD, USA). Western blot and densitometry analysis for total beta-catenin levels were performed for three experimental replicates.

### Real-time quantitative RT-PCR

Real-time PCR amplification was performed using the Applied Biosystems StepOne RT-PCR system (Applied Biosystems, Burlington, ON, Canada). Kidney tissue was lysed by tituration using a 25-gauge needle in RNA lysis buffer and RNA was isolated using the RNeasy Micro kit (QIAGEN, Germantown, MD, USA). RNA was reverse transcribed to cDNA using the $RT^2$ First Strand kit (QIAGEN, Germantown, MD. USA). Real-time PCR reaction mix contained 3 ng of each cDNA sample, 300 nM of primer, 1X SYBR Green ROX PCR Master Mix (QIAGEN, Germantown, MD, USA) and RNAse-free $H_2O$ to a total volume of 25μL per reaction. Relative mRNA levels were quantitated using the $2^{\Delta\Delta Ct}$ quantitation method, using cDNA from the untreated kidneys as reference sample and *B2M* as endogenous control. Primers used were *Pax2* (forward: 5′-`taggaaggacgctcaaagactc`-3′; reverse 5′-`taatggagactc ccagagtggt`-3′), *Six2* (forward 5′-`cttctcatcctcggaactgc`-3′; reverse: 5′-`gga gaacagcgagaactcca`-3′), *Gdnf* (forward: 5′- `agccctgaacatattgtcacct`-3′; reverse: 5′-`tcccctatgttctcctgtctgt`-3′), *B2M* (forward: 5′-`ccgaacatactgaa ctgctacg`-3′; reverse: 5′- `cacatgtctcgatccagtaga`-3′).

### Quantifying branching morphogenesis and self-renewing cells

Branching morphogenesis was quantitated by counting the number of ureteric branch tips on the outer surface of the kidney. At least six images of whole-mount cytokeratin stained kidneys per treatment group were used for the quantitation. The number of self-renewing cells were quantitated by counting the number of Six2-positive cells in the condensed mesenchyme, and the total area of the condensed mesenchyme per image was measured using ImageJ. Two sections 10μm apart from at least 6 kidneys from each treatment group were counted.

### Quantifying nephrogenic structures

H&E-stained or cytokeratin and Pax2 immuno-stained kidney tissue were imaged and the area of each kidney tissue was measured using ImageJ. Developing nephron structures (comma-shaped, S-shaped, and maturing glomeruli) were counted per image. Two sections 10μm apart from at least 5 kidneys from each treatment group were counted.

### Statistical analysis

All statistical analyses were performed using GraphPad Prism software (version 8.1.2, San Diego, CA, USA). A one-way ANOVA followed by a Tukey's post-hoc test was performed to analyze changes in branching morphogenesis, number of self-renewing cells, total beta-catenin levels and target gene expression levels in quercetin treated wild type kidneys compared to untreated controls. Two-tailed Student's t-test was performed to analyze changes in the number of nephrogenic structures and target gene expression levels in quercetin treated dysplastic kidneys compared to untreated dysplastic kidneys. In all analyses, p-values of <0.05 were considered statistically significant and data are reported as Mean ± SEM.

## Results

### Quercetin alters branching morphogenesis and nephrogenesis

Our previous studies have demonstrated that beta-catenin overexpression contributes to severe renal dysplasia in both the human condition and mouse model [14–16] Therefore, identifying factors that can reduce beta-catenin can contribute to improvements in the severity of renal dysplasia. Quercetin has been shown to reduce beta-catenin activity in postnatal models of various cancers [21, 30, 31] and kidney fibrosis [20]. However, the effects of quercetin on beta-catenin during organ development and developmental abnormalities is not known. We first determined whether quercetin has any effects on normal kidney development. To do this, we resected kidneys from E13.5 wild type mouse embryos. The kidneys were cultured in the absence or presence of 40μM, 80μM, or 160μM of quercetin. These concentrations were previously shown to alter beta-catenin cellular distribution and transcriptional activity in cancer cells [21, 30, 31] and in postnatal kidney tissue [20]. After 48 hours of organ culture, we performed whole-mount immunofluorescence to visualize branching morphogenesis (Fig 1A–1D). In the absence of quercetin treatment, cytokeratin staining demonstrated the typical well-organized branch pattern with numerous branch tips (Fig 1A, asterisk). In contrast, a dose-dependent reduction in branching morphogenesis was observed in the quercetin-treated kidneys (Fig 1B, 1C and 1D). The quantitation of the ureteric branch tips confirmed a 1.4-fold (0μM quercetin: 93.57±8.641, n = 7 vs. 40μM quercetin: 64.83±8.072, n = 6, p = 0.0275 (*)), 1.9-fold (0μM quercetin: 93.57±8.641, n = 7 vs. 80μM quercetin: 50.22±5.096, n = 9, p = 0.0002 (**)), and 3.9-fold (0μM quercetin: 93.57±8.641, n = 7 vs. 160μM quercetin: 23.88±3.507, n = 8, p<0.0001 [***])) reduction in the number of ureteric branch tips (Fig 1E). We also noted that the increasing dose of quercetin resulted in changes in the integrity of the ureteric epithelium. In untreated samples, the epithelium was a highly organized columnar epithelium layer (Fig 1A inset). The quercetin treated kidneys demonstrated a disorganized cuboidal epithelium that lacked continuity (Fig 1B–1D insets). Of note, the epithelium looked more organized and adherent to neighboring cells in the 160μM treated kidneys when compared to the 80μM treatment.

We next analyzed the effects of quercetin treatment on nephron formation. In untreated kidneys, the Pax2+ nephron progenitors were organized within the nephrogenic zone (Fig 1F, dotted line) and several Pax2+ developing and maturing nephrons were observed throughout the kidney tissue (Fig 1F, arrows). In contrast, kidneys treated with 40μM quercetin showed a reduced and disorganized Pax2+ nephron progenitor population, an abnormally expanded nephrogenic zone (Fig 1G, dotted line), and several Pax2+ underdeveloped nephrogenic structures (Fig 1G, arrows). These abnormalities were further exacerbated with 80μM quercetin and 160μM quercetin (Fig 1H–1I). Six2 immunofluorescence confirmed the reductions (Fig 1N) and disorganization in the nephron progenitor cell population (Fig 1J–1M). We also consistently observed that increasing the quercetin dose resulted in a gap between the nephron progenitors and ureteric epithelium (Fig 1K–1M double arrowhead). Altogether, these findings demonstrate, for the first time, that quercetin treatment causes a dose-dependent disruption in kidney development by reducing branching morphogenesis, the nephron progenitor population, and nephrogenesis during normal kidney development.

### Quercetin decreases nuclear beta-catenin levels and transcriptional activity

The phenotypes observed in the quercetin treated kidneys were similar to kidneys from mice with beta-catenin deficiency in the ureteric epithelium or mesenchyme [14, 34, 35]. Therefore, we next analyzed beta-catenin expression and transcriptional activity on E13.5 wild type

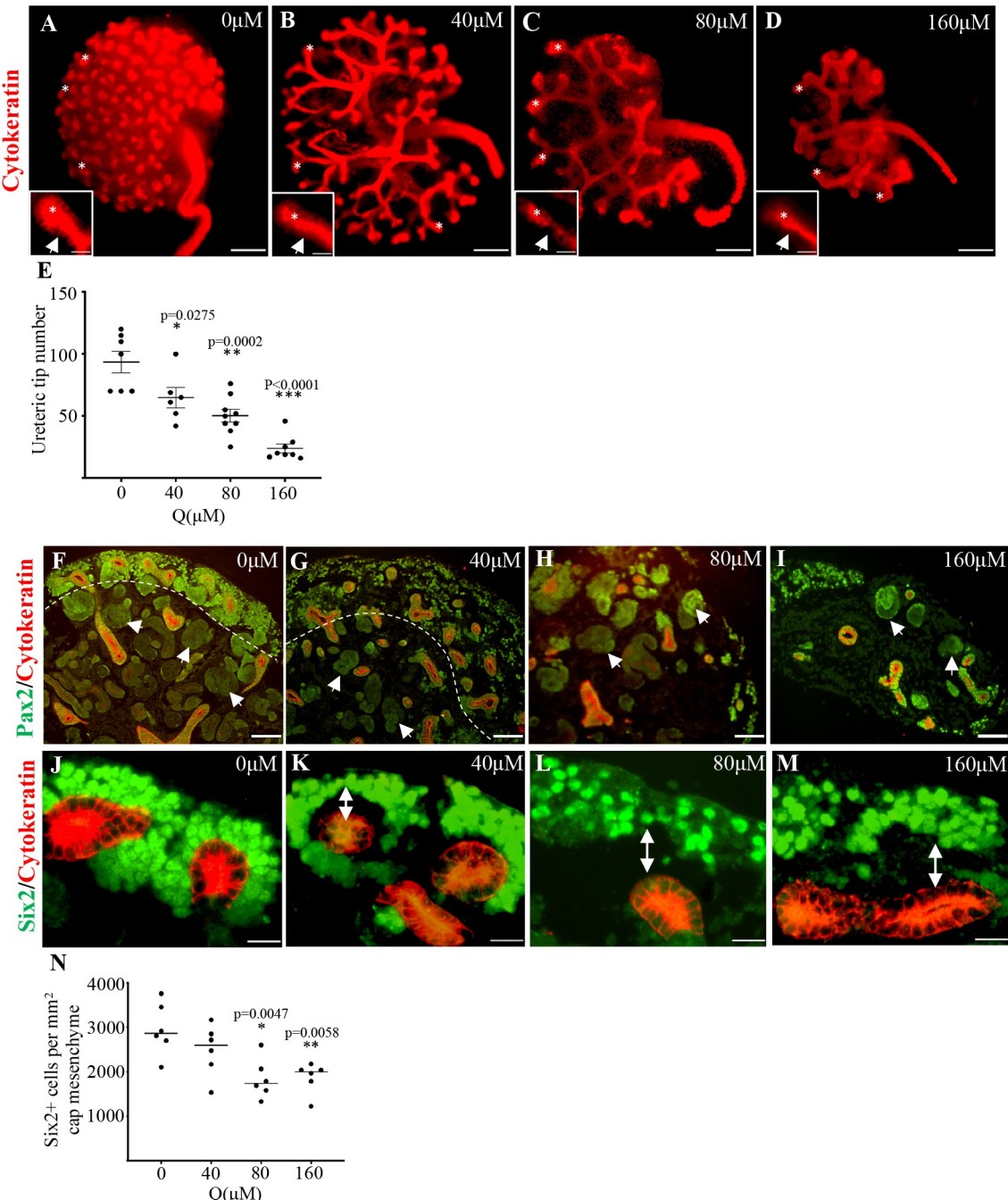

**Fig 1. Quercetin treatment reduces branching morphogenesis in wild type embryonic kidneys.** (**A-D**) Cytokeratin immunofluorescence in the presence or absence of quercetin. The 40μM and 80μM quercetin treated kidneys exhibit reduced ureteric branch tips (*) and increased spacing between branch stalks and tips. The 160μM dose demonstrates very few short and thick ureteric branches associated with a disorganized branch pattern. The insets highlight a worsening ureteric epithelial organization as demonstrated poorly aligned epithelial nuclei (arrow). (**E**) Dot plot demonstrating the quantification of ureteric branch tips (p = 0.0275 (*), p = 0.0002 (**), p<0.0001 (***)). (**F-I**) Pax2/cytokeratin co-immunofluorescence. Untreated kidneys show a distinct nephrogenic zone (dotted lines) with several Pax2+ cells around cytokeratin+ ureteric tips. Several developing nephrons are observed (arrows). The 40μM, 80μM, 160μM quercetin shows a reduced number of Pax2+ cells, ill-defined nephrogenic zone, and very few normal developing nephrons. (**J-M**) Six2/cytokeratin co-immunofluorescence. Untreated kidneys show Six2+ cells tightly clustered around the ureteric tips. The 40μM, 80μM, 160μM quercetin shows a progressive reduction in Six2+ cells. The Six2+ cells in the 80μM and 160μM dose do not cluster around the ureteric epithelium. Note the increased distance of the Six2+ positive cells from the ureteric epithelium with increasing quercetin dose. (**N**) Quantification of Six2+ cells in untreated and quercetin treated kidneys (p = 0.0047(*), p = 0.0058 (**)). Scale bars = 200μm (A-D) (insets = 20μm), 100μm (E-H), 20μm (I-L).

kidneys grown in the presence or absence of quercetin. Initially, we investigated beta-catenin levels by performing immunoblotting. This analysis demonstrated no changes in the overall cellular expression level of beta-catenin (Fig 2A and 2B). However, the analysis of beta-catenin by immunohistochemistry demonstrated that increasing doses of quercetin resulted in alterations in the intracellular distribution of beta-catenin expression. In untreated kidney samples beta-catenin was predominantly expressed in the nucleus of the nephron progenitors (Fig 2C and 2C' arrows). While, in the ureteric epithelium, beta-catenin localized to the nucleus, cytoplasm, and membrane (Fig 2C and 2C' arrowheads). Treatment with increasing doses of quercetin resulted in a progressive reduction of beta-catenin in the nucleus (Fig 2D', 2E' and 2F' arrows) and increasing expression in the cytoplasm and membrane of both the mesenchyme and ureteric epithelium (Fig 2D', 2E' and 2F' arrowheads). We next determined if the reductions in nuclear beta-catenin expression also resulted in changes to the expression levels of beta-catenin target genes *Pax2*, *Six2* and *Gdnf*. Quantitative RT-PCR demonstrated that quercetin treated kidneys had a dose-dependent reduction in *Pax2* expression (Fig 2G) (0µM quercetin: 1.00, vs. 40µM quercetin: $0.6207 \pm 0.09948$, $p = 0.0151$ (*); 80µM quercetin: $0.5071 \pm 0.05380$, $p = 0.0033$ (**); 160µM quercetin: $0.3261 \pm 0.06756$, $p = 0.0004$ (***), n = 3). The levels of *Six2* (Fig 2H) and *Gdnf* (Fig 2I) expression in quercetin treated kidneys also demonstrated reductions compared to untreated controls, although not in a dose dependent manner. Taken together, these findings demonstrate that quercetin does not change the total expression levels of beta-catenin, but rather results in a progressive redistribution of beta-catenin from the nucleus to the cell membrane.

## Quercetin treatment reduces the severity of renal dysplasia

Previous work from our lab investigated the consequences of beta-catenin overexpression in renal dysplasia [14]. Our work demonstrated that transgenic mouse models that overexpress beta-catenin result in the dysregulation of the normal genetic programs required for kidney development and as a result contributes to the genesis of renal dysplasia [14–16]. Therefore, we next investigated if quercetin can improve the severity of renal dysplasia. Mice with renal dysplasia were generated by crossing *Rarb2Cre* male mice (Cre expression limited to the metanephric mesenchyme) with female mice containing *LoxP* sites flanking exon 3 of the beta-catenin allele [14]. This cross removes phosphorylation sites in beta-catenin that are required for its degradation. Since these mutant mice have bilateral renal dysplasia caused by beta-catenin, we termed these mutant mice <u>r</u>enal <u>d</u>ysplasia by <u>b</u>eta-catenin ($RD^B$). $RD^B$ mice exhibit similar histopathological changes in the left and right kidneys and among littermates and in different litters. $RD^B$ mutant kidneys were resected at E13.5 and cultured for 48 hours in the presence or absence of 40µM quercetin. The 40µM quercetin dose was used since this dose did not have severe effects on branching morphogenesis and nephrogenesis. It also altered beta-catenin intracellular distribution but not its total expression levels in wild type kidneys. In the absence of quercetin, H&E staining demonstrated that the $RD^B$ kidney architecture was disorganized and lacked a distinct nephrogenic zone (Fig 3A and 3B). The nephron progenitor population was sparse (Fig 3B, arrow), there were numerous clusters of undifferentiated mesenchyme cells (Fig 3B, asterisks), and an abnormal dilated ureteric epithelium was observed (Fig 3B, arrowhead). Very few developing nephrons were observed, and several poorly formed glomeruli were detected (Fig 3B, block arrows). The disorganization was highlighted with Pax2 staining which demonstrated very few Pax2+ cells in the outer cortex. Several individual Pax2 + cells were observed deeper in the kidney tissue (Fig 3C, arrow), again highlighting the disorganization. Almost all developing tubules were cytokeratin positive, indicating that these structures are disorganized ureteric epithelium (Fig 3C, arrowhead). The analysis of the Six2

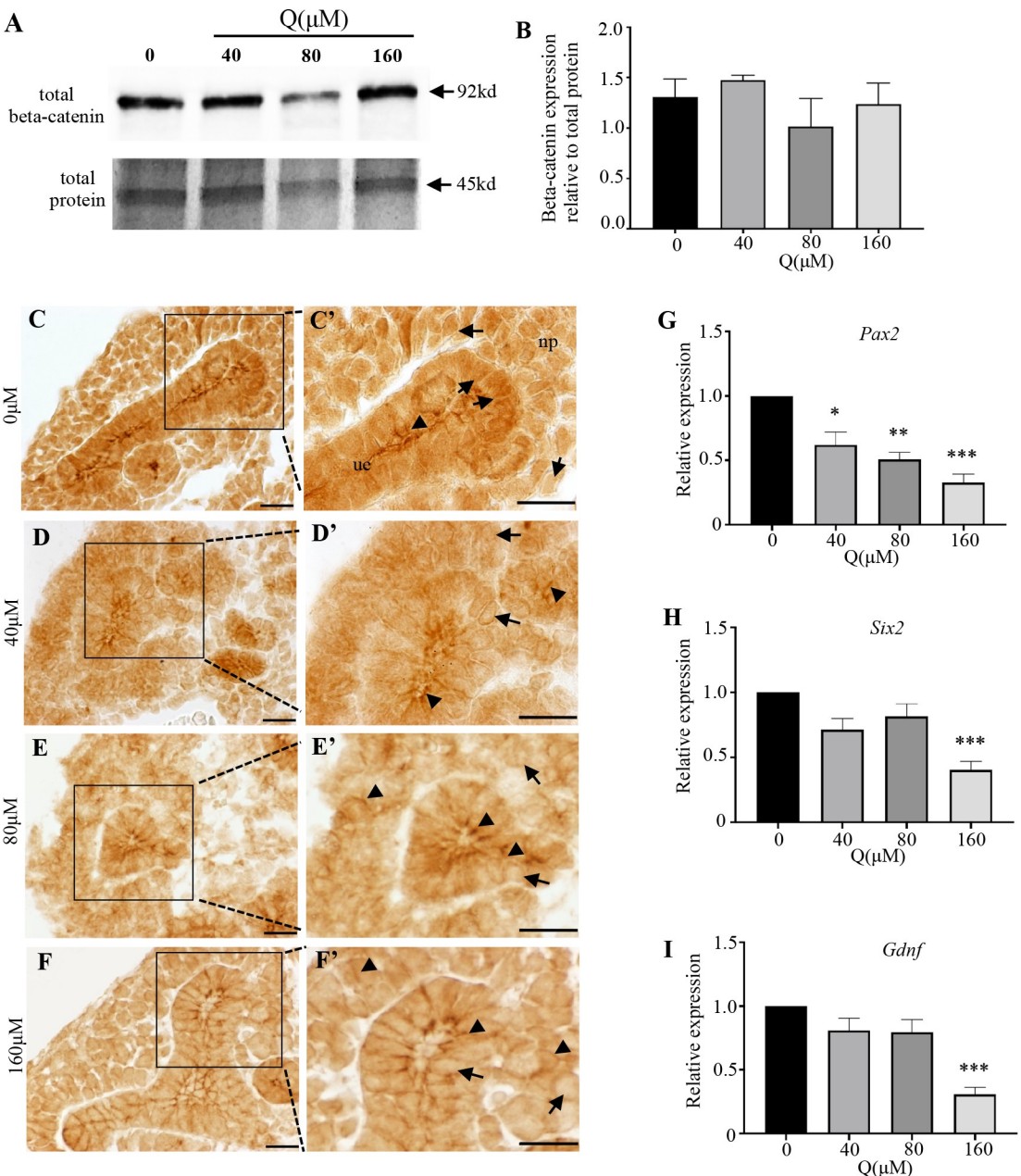

**Fig 2. Quercetin treatment alters the intracellular distribution of beta-catenin in wild type embryonic kidneys. (A)** Western blot analysis for beta-catenin on untreated and quercetin treated kidneys. Total protein was used as loading control. **(B)** Densitometry analysis demonstrating no changes in total beta-catenin expression levels. Mean relative expression levels shown are taken from three experimental replicates. **(C-F and C'-F')** Beta-catenin immunohistochemistry demonstrating increased quercetin dose results in a progressive reduction in nuclear beta-catenin (arrow) and more robust expression in the membrane (arrowhead) in the mesenchyme (M) and ureteric epithelium (UE). C'-F' are magnified images of the boxed regions. Scale bars = 20μm (C-F), 10μm (C'-F'). **(G-I)** Quantitative RT-PCR for *Pax2*, *Six2* and *Gdnf*. Quercetin treatment results in a dose-dependent decrease in *Pax2* (p = 0.0151 (*), p = 0.0033 (**), p = 0.0004 (***)) and reductions in *Six2* (p = 0.0019 (***)) and *Gdnf* (p = 0.0008 (***)) only at the 160μM dose.

+ nephron progenitor population demonstrated clusters around the cytokeratin+ ureteric tips that are about 4–6 cells deep. These clusters of Six2+ cells were sporadically and sparsely distributed throughout the outer cortex (Fig 3D). In contrast to untreated $RD^B$ kidneys, the

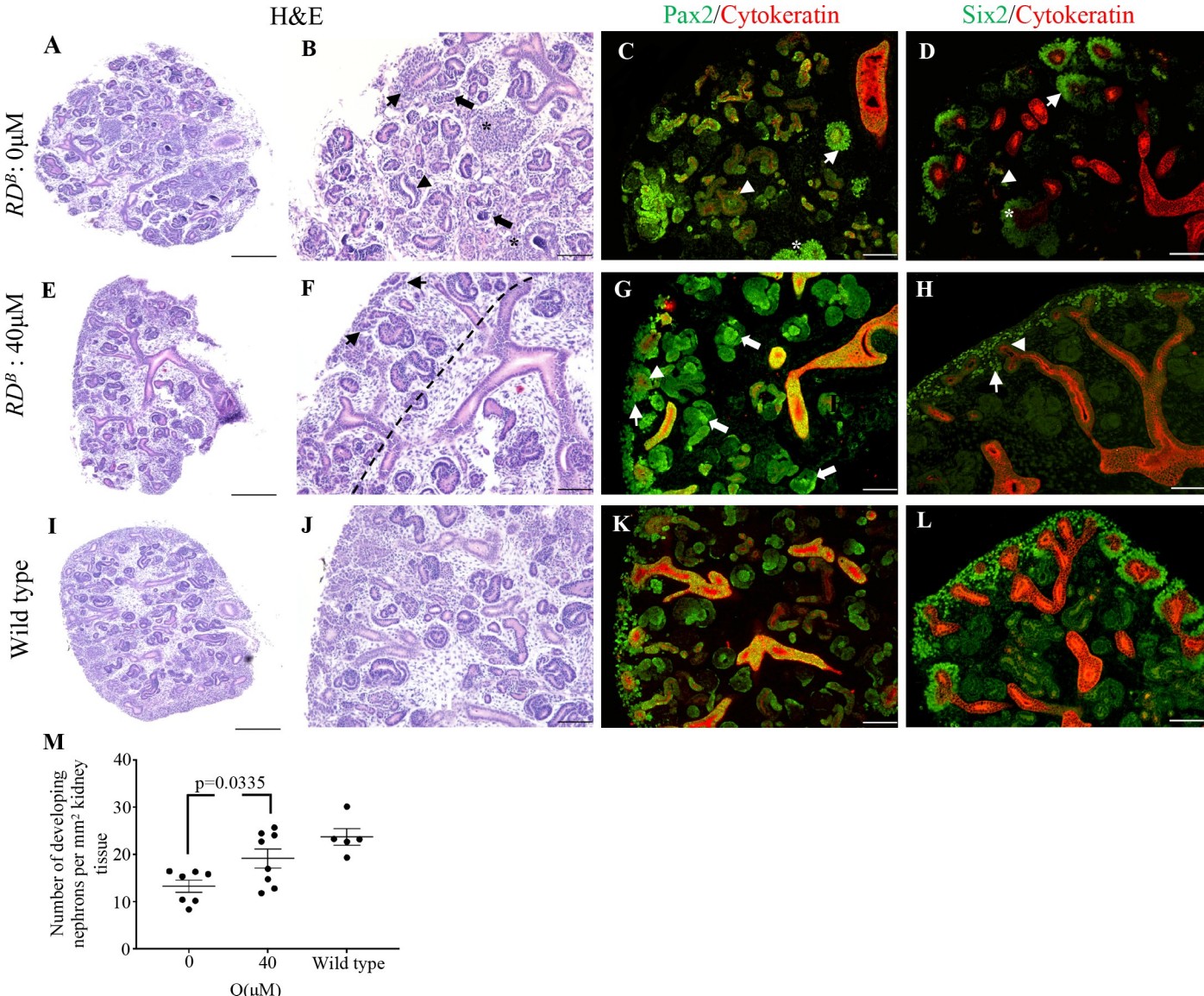

**Fig 3. Quercetin reduces the severity of renal dysplasia.** (**A-L**) H&E staining, Pax2/cytokeratin, and Six2/cytokeratin co-immunofluorescence in untreated and treated dysplastic kidneys. (**A-D**) Untreated kidneys lack a distinct nephrogenic zone and have abnormal nephron progenitor formation (arrow), clusters of undifferentiated mesenchyme cells (asterisk), abnormal ureteric epithelium (arrowhead), and very few developing nephrons. (**E-H**) Dysplastic kidneys treated with 40μM quercetin show a distinct nephrogenic zone (dotted line) containing nephron progenitors (arrow) surrounding ureteric epithelium (arrowhead) and several developing nephrogenic structures (block arrows). (**I-L**) Wild type kidneys are shown for comparison. (**M**) Quantification of developing nephron structures in untreated and treated dysplastic kidneys. Dysplastic kidneys treated with 40μM have increased nephrogenic structures compared to untreated controls (p = 0.0335). The number of developing nephrons in wild type kidneys is shown for comparison. Scale bars = 100μm.

histology in the quercetin treated kidneys demonstrated an overall improvement in kidney architecture (Fig 3E and 3F). The treated $RD^B$ kidneys exhibited a distinct nephrogenic zone (Fig 3F, dotted line) and contained several Pax2+ cells (Fig 3G, arrow) and developing nephrons (Fig 3G, block arrows). Cytokeratin negative tubules were found deeper in the kidney parenchyma indicating the presence of several developing and maturing nephrons (Fig 3G, block arrows). The cytokeratin+ structures exhibited a distinct organization (Fig 3G and 3H). In addition, the Six2+ nephron progenitors were observed in a consistent pattern around the

ureteric bud tips (Fig 3H, arrow). The kidney phenotype and pattern of nephrogenesis in quercetin treated dysplastic kidneys was more similar to wild type kidneys (Fig 3I–3L).

We next determined if the improvements in nephrogenesis and branching morphogenesis translated to an increase in the number of developing nephrons. Quercetin treatment of $RD^B$ kidneys resulted in a 1.45-fold increase in the number of nephrogenic structures when compared to untreated dysplastic kidneys (untreated: 13.25±1.309 n = 7 vs. 40μM quercetin: 19.14 ±2.009 n = 8, p = 0.0335; wild type: 23.71±1.757 n = 5) (Fig 3M). While this number was less than that observed in wild type kidneys, this did represent a significant improvement over the untreated samples. Altogether, these findings demonstrate that quercetin treatment of $RD^B$ kidneys resulted in a significant improvement in kidney architecture, nephrogenesis, and organization of ureteric branches. However, quercetin treatment does not result in a complete rescue of renal dysplasia.

## Quercetin treatment reduces nuclear beta-catenin levels and transcriptional activity in $RD^B$ dysplastic kidneys

We next examined if the reduced severity of renal dysplasia is associated with changes in beta-catenin expression. In the untreated $RD^B$ renal dysplasia mouse model, beta-catenin was over-expressed in the clusters of undifferentiated mesenchyme that were found sporadically distributed throughout the cortex and medulla (Fig 4A and 4B). In these mesenchymal clusters, beta-catenin expression was observed in the nucleus and cytoplasm with very little expression observed in the membrane (Fig 4A' and 4B' arrows). Of note, to visualize the expression pattern in these clusters, the DAB staining time was markedly reduced. This makes the beta-catenin expression in the surrounding tissue lighter. In contrast to the untreated $RD^B$ kidneys, quercetin treated kidneys did not exhibit the mesenchyme clusters that overexpress beta-catenin (Fig 4C and 4D). The mesenchyme cells around the ureteric tips (Fig 4C', arrows) demonstrated beta-catenin levels similar to that observed in wild type (Fig 4E and 4F). In the quercetin treated $RD^B$ kidneys, several developing and maturing nephrons were found that express beta-catenin in a pattern similar to wild types (Fig 4D' and 4F'). We next analyzed the expression of the beta-catenin target genes *Pax2*, *Six2*, and *Gdnf*. Quantitative RT-PCR demonstrated a reduction in *Pax2* expression in quercetin treated $RD^B$ kidneys (untreated: 1.00 vs. 40μM quercetin: 0.6853±0.06944, n = 5, p = 0.0106 (*)) (Fig 4G). However, no changes were observed in *Six2* or *Gdnf* (Fig 4H and 4I). Taken together, these findings demonstrate that quercetin treatment reduces the beta-catenin overexpression in the $RD^B$ mutant model without having significant effects on the beta-catenin expression in the surrounding tissues.

## Quercetin improves epithelial organization in $RD^B$ dysplastic kidneys

We next determined how quercetin treatment results in the improvement in nephrogenesis and overall histopathology in renal dysplasia. High nuclear vimentin levels are associated with de-differentiation of epithelial cells or maintaining a mesenchymal cell fate [36]. Therefore, nuclear vimentin may prevent epithelial transition and contribute to the reduction in developing nephrons and the formation of stalled nephrogenic structures. Immunohistochemistry using vimentin demonstrated a strong and predominant nuclear expression of vimentin in the clusters of uninduced mesenchyme cells in the untreated dysplastic kidneys (Fig 5A, arrows and inset). However, in the quercetin treated dysplastic kidneys, the condensing mesenchymal cells around the ureteric tips demonstrated a more prominent vimentin expression in the cytoplasm and less intense in the nucleus (Fig 5B, arrow and inset). This pattern was more similar to that observed in wild type kidneys (Fig 5C, arrow and inset). Considering vimentin is a mesenchymal marker, these results suggest that quercetin treatment may facilitate mesenchymal

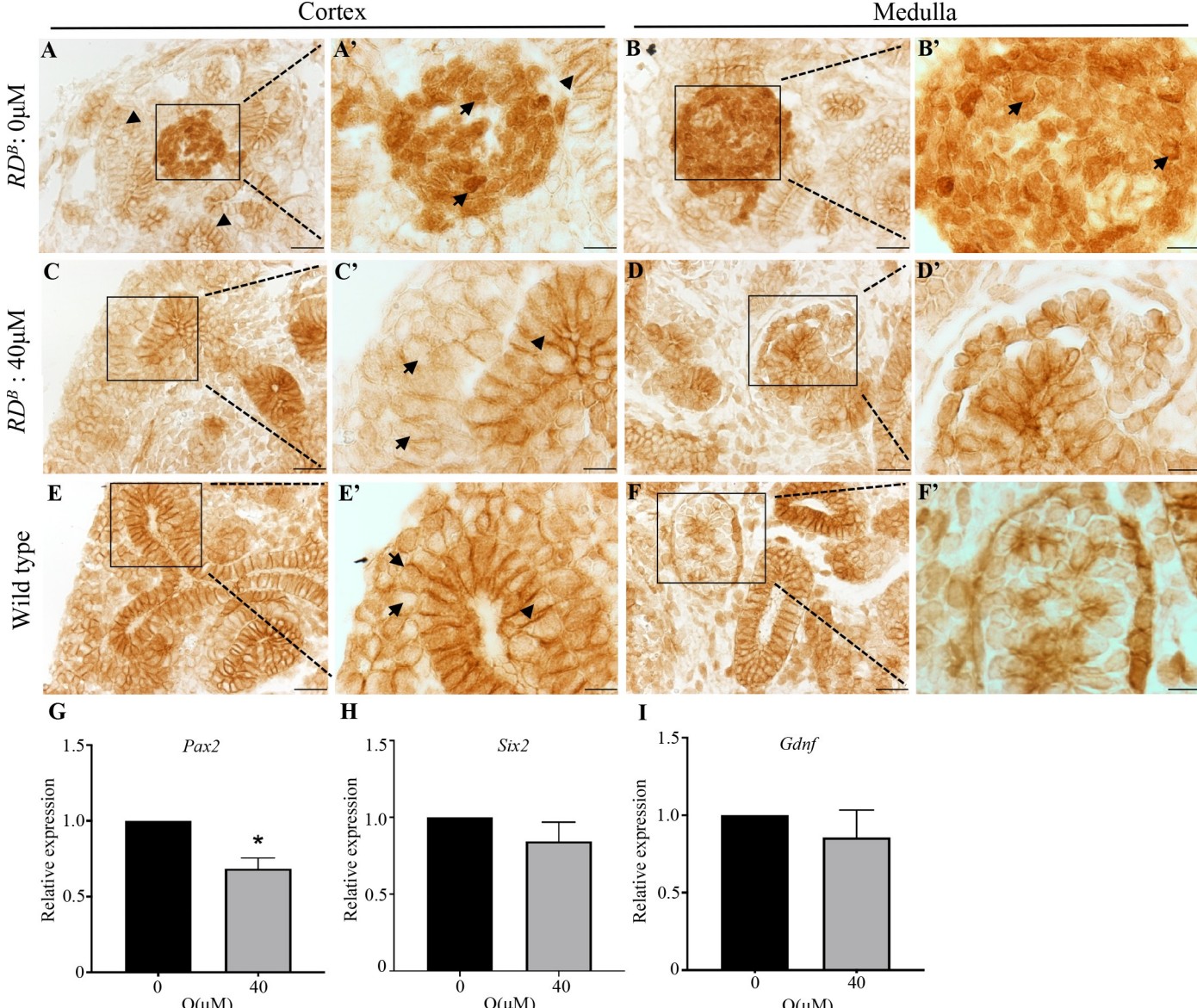

**Fig 4. Quercetin treatment in dysplastic kidneys reduces nuclear beta-catenin accumulation.** (A-F) Beta-catenin immunohistochemistry in untreated and treated dysplastic and wild type kidneys; (A'-F') are high power insets of boxed regions. (A,B) Untreated dysplastic kidneys display prominent nuclear beta-catenin (arrow) in clusters of mesenchyme in the cortex and medulla. Beta-catenin is unchanged in the ureteric epithelium (arrowhead). (C,D) Dysplastic kidneys treated with 40μM quercetin do not show the mesenchyme clusters. The cap mesenchyme demonstrates decreased nuclear beta-catenin (arrow) compared to untreated. Developing and maturing nephrons in the medulla display normal beta-catenin expression. (E,F) Beta-catenin in wild type kidneys is shown for comparison. Scale bars = 20μm (A-F), 10μm (A'-F'). (G-I) qRT-PCR for *Pax2*, *Six2* and *Gdnf*. Quercetin treatment results in reduced *Pax2* expression (p = 0.0106 (*)). *Six2* and *Gdnf* remained unchanged.

to epithelial transition. In normally developing kidneys, NCAM localizes to the membrane of cells transitioning to an epithelial cell fate (such as the condensed mesenchyme cells, renal vesicles, comma-shaped structures and S-shaped structures) [37]. The proper localization of NCAM and E-cadherin to the cell membrane is necessary for the proper formation of the nephrogenic epithelium during nephrogenesis [37, 38]. Therefore, we next analyzed NCAM and E-cadherin expression. In the untreated dysplastic kidneys, NCAM lacked membrane localization in the cap mesenchyme and appeared more cytoplasmic (Fig 5D, arrows). In the

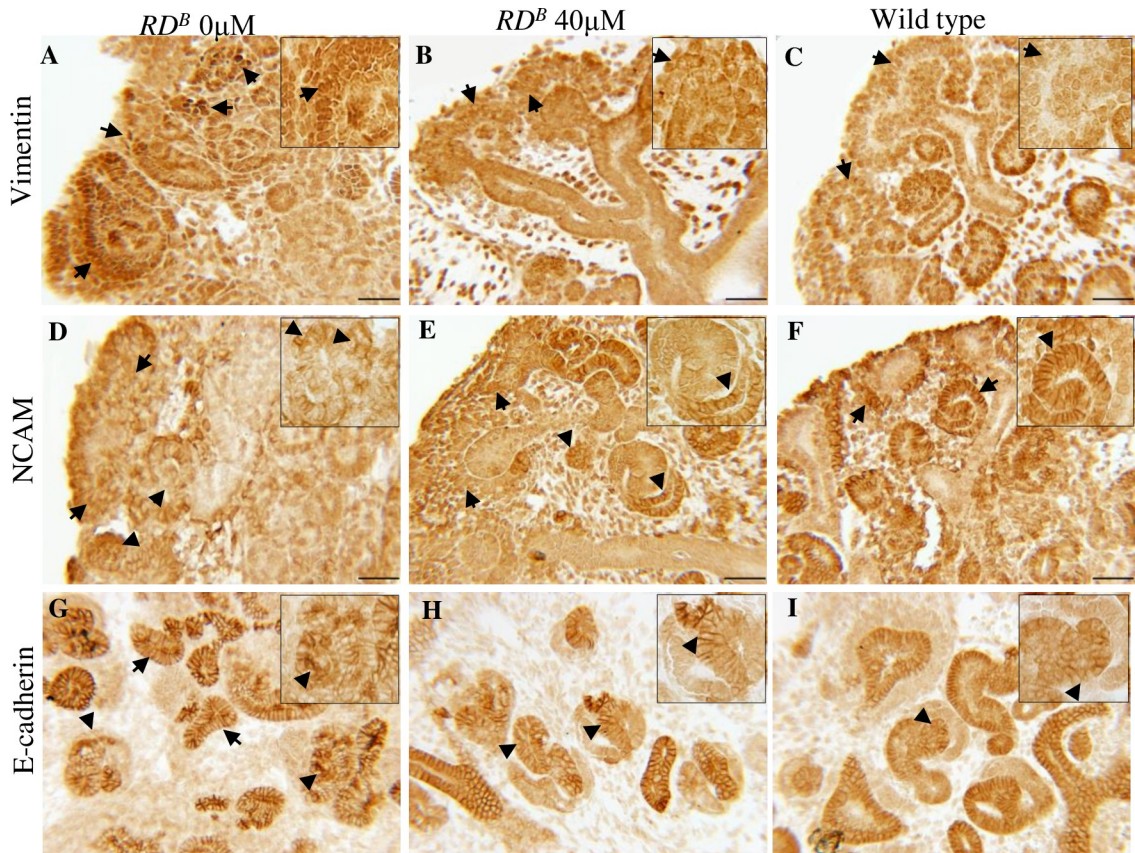

**Fig 5. Quercetin improves mesenchyme and epithelial organization in dysplastic kidneys. (A-I)** Vimentin, NCAM, and E-cadherin immunohistochemistry in treated and untreated dysplastic kidneys and wild type kidneys. **(A-C)** Vimentin expression in untreated dysplastic kidneys demonstrates nuclear expression in clusters of disorganized mesenchyme (arrow). Dysplastic kidneys treated with quercetin have less nuclear and more cytoplasmic vimentin expression. This pattern is more similar to wild type kidneys. **(D-F)** NCAM expression in untreated dysplastic kidneys lacks intracellular organization, is weakly expressed in the mesenchyme and is sporadically expressed in the membranes of developing nephrons (arrow). Quercetin-treated dysplastic kidneys show a more organized membrane expression in developing nephrons. Wild type kidneys are shown for comparison. **(G-I)** E-cadherin expression in untreated dysplastic kidneys shows abnormally developing nephrons with E-cadherin localized in the cytoplasm and sporadically at the cell membrane (arrow). Quercetin treated dysplastic kidneys show improved E-cadherin expression primarily restricted to the cell membrane in developing nephrons (arrow). Wild type kidneys are shown for comparison. Scale bars = 50μm.

abnormally developing nephron structures, the membrane expression was sporadic (Fig 5D, arrowhead) and was often observed in the cytoplasm (Fig 5D inset). In the quercetin treated dysplastic kidneys, NCAM expression was reduced in the cap mesenchyme and was not expressed in the cell membrane (Fig 5E, arrow). However, in the early developing nephrogenic structures, a more distinct and consistent membrane expression pattern was observed (Fig 5E, arrowhead). The pattern observed in nephrogenic structures in the quercetin treated kidneys was similar to that observed in wild type (Fig 5F, arrow and inset).

Our results demonstrated that quercetin treatment resulted in beta-catenin expression that is more prominently expressed at the cell membrane (Fig 2C–2F). When beta-catenin localizes to the cell membrane, it supports the formation of cell-cell adhesion through E-cadherin junctions [8, 39]. Untreated dysplastic kidneys demonstrated E-cadherin expression primarily in the membrane of the ureteric epithelium (Fig 5G, arrow). However, in the abnormally developing nephrons, E-cadherin expression was observed at low levels in the membrane and cytoplasm (Fig 5G, arrowhead and inset). In contrast, developing nephrons in the quercetin

treated dysplastic kidneys demonstrated a more prominent membranous pattern of E-cadherin expression between epithelial cells that was limited to specific segments of the developing nephron (Fig 5H, arrowhead and inset). The pattern observed in the quercetin treated kidneys was similar to wild type (Fig 5I, arrowhead and inset). These results demonstrate that quercetin treatment of $RD^B$ dysplastic kidneys results in an improvement of the developing nephron epithelia.

## Human renal dysplasia demonstrates disorganized vimentin and e-cadherin expression

We next determined if the observations of disrupted vimentin and E-cadherin expression in our mouse model is consistent in human renal dysplasia. In normal kidney tissue, low levels of vimentin expression is detected in glomeruli and renal tubules (Fig 6A and 6A'-arrows). The human dysplastic kidney tissue revealed abnormally developing glomeruli with dilated tubules that are surrounded by undifferentiated mesenchymal cells. The dilated tubular epithelium did not express vimentin (Fig 6B-arrowhead). However, the cells surrounding the tubules showed abnormally high intracellular vimentin expression (Fig 6B and 6B'-arrows). We next analyzed E-cadherin expression. In normal kidney tissue E-cadherin was detected in the basolateral membranes and between proximal tubule epithelial cells (Fig 6C and 6C'-arrows). In renal dysplasia, the epithelial tubules lacked membrane E-cadherin expression and instead showed a more cytoplasmic staining pattern (Fig 6D and 6D'-arrows). In some epithelial cells E-cadherin expression was completely absent (Fig 6D-arrowhead). Taken together, these findings confirm that abnormalities in the expression of vimentin and E-cadherin are also a feature in human renal dysplasia.

## Discussion

Renal dysplasia results from the abnormal development of the kidney and is the most common cause of childhood renal failure. Despite this, there is no cure. Our previous work demonstrated that some cases of human renal dysplasia are associated with elevated levels of nuclear beta-catenin [14–16, 40]. This leads to a disrupted expression of genes essential for kidney development, leading to the pathogenesis of renal dysplasia. Therefore, targeting the excessive nuclear beta-catenin can be a potential therapeutic intervention for renal dysplasia. Here, we demonstrate that quercetin reduced nuclear beta-catenin levels and its transcription of kidney development genes. As expected, this resulted in disruptions in branching morphogenesis and nephrogenesis. Further, using an established mouse model of renal dysplasia, we demonstrated that dysplastic kidneys treated with quercetin reduced the severity of renal dysplasia by reducing the elevated nuclear beta-catenin levels. Consequently, this resulted in a more intact developing nephron epithelia which resulted in an increase in the number of properly forming nephrons. Taken together, our data demonstrates that quercetin treatment results in a partial rescue of renal dysplasia.

The reductions in branching morphogenesis and nephrogenesis observed in quercetin treated wild type kidneys is consistent with mutant mice that lack beta-catenin in the ureteric epithelium or mesenchyme-derived structures. The deletion of beta-catenin specifically in the condensed mesenchyme cells or in renal vesicles results in a decrease in nephrogenesis and associated reductions in *Pax2* and *Six2* expression, which is identical to our results [35]. Further, our analysis of branching morphogenesis in quercetin treated kidneys demonstrates a significant decrease in the number of ureteric branch tips and an associated reduction in *Gdnf* expression. These results are consistent with the deletion of beta-catenin specifically in the ureteric epithelium which demonstrate a premature cessation of ureteric branching and reduced

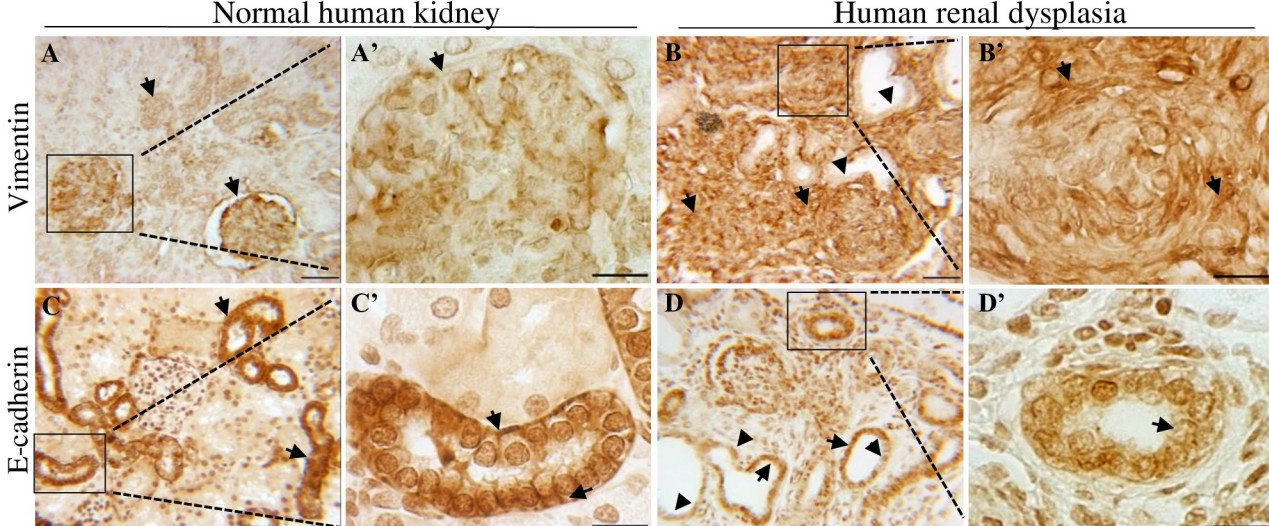

**Fig 6. Human renal dysplasia demonstrates abnormal mesenchyme and epithelial organization. (A-D)** Vimentin and E-cadherin immunohistochemistry in normal and dysplastic human kidney tissue; **(A'-D')** are high power insets of boxed regions. **(A,A')** In normal human tissue low levels of vimentin expression (arrow) are observed in the glomeruli and renal tubules. **(B,B')** Dysplastic kidney tissue has high vimentin levels in mesenchymal cells (arrow) surrounding dilated tubules (arrowhead). **(C,C')** In normal kidney tissue E-cadherin is strongly expressed in the basolateral cell membrane of the epithelium tubules (arrow). **(D,D')** In dysplastic kidney tissue E-cadherin is absent from the membrane and weakly expressed in the cytoplasm of epithelial tubules (arrow). In some epithelial cells E-cadherin is completely absent (arrowhead). Scale bars = 50µm (A-D), 10µm (A'-D').

expression of *Gdnf* [34, 41]. Given that our quercetin treated wild type kidneys demonstrated phenotypic similarities to *in vivo* mouse models with a targeted deletion of beta-catenin, our findings support that quercetin affects kidney development in a beta-catenin dependent manner.

We observed that quercetin decreases nuclear beta-catenin during normal kidney development and in renal dysplasia. Our findings are similar to that observed in breast, colon, and pancreatic cancer cells [21, 30, 31]. In these models, quercetin treatment reduced abnormal nuclear beta-catenin levels and reduced the transcription of beta-catenin target genes *c-myc* and *cyclin D1* [21, 30, 31]. This resulted in decreased cancer cell survival, proliferation, and invasiveness. Further, in the kidney, quercetin is also effective in reducing kidney damage resulting from abnormal beta-catenin signaling. Increased levels of nuclear beta-catenin and signaling activity has also been shown in patients with chronic kidney disease [42]. In the presence of kidney injury, elevated beta-catenin levels lead to fibrosis due to the increased activation of various beta-catenin dependent pathways involved in the fibrotic response, such as the Wnt signaling pathway, the renin-angiotensin system, and Tgfb1/Smad pathway. This results in the upregulation of pro-fibrotic factors such as collagen I, fibronectin, Snail, Twist and MMP-7 and abnormally increased activity of renal interstitial fibroblasts, eventually leading to chronic kidney damage [43–46]. A study has shown that mice with acute kidney injury demonstrate an increase in nuclear beta-catenin in interstitial fibroblast cells, and quercetin treatment resulted in reduced nuclear beta-catenin levels and reduced expression of pro-fibrotic factors fibronectin, α-smooth muscle actin, Snail and Twist, leading to a reduced severity of renal fibrosis [20]. These studies provide further support that reducing abnormal beta-catenin levels can lead to reduced severity of kidney damage. However, these previous studies were performed in postnatal mice. Our studies in developing kidneys extend these studies and demonstrate that quercetin treatment is also associated with reductions in nuclear beta-catenin

and gene expression during kidney development. Specifically, *Pax2* demonstrated dose dependent reductions with quercetin treatment. During mesenchyme to epithelial transition Pax2 becomes gradually downregulated and restricted to specific segments of the maturing nephron [47]. If Pax2 is ectopically expressed it leads to sustained mesenchyme proliferation and contributes to the formation of stalled nephrons by preventing further differentiation of the nephron segments [47]. This suggests that quercetin treatment decreases the elevated *Pax2* expression which results in improved nephrogenesis.

In normally developing kidneys, quercetin resulted in a dose dependent reduction in the nuclear expression of beta-catenin with a corresponding accumulation in the membrane-bound compartment. This suggests that quercetin prevents beta-catenin from entering the nucleus or assists in its exit from the nucleus. While it is unknown how beta-catenin enters and exits the nucleus, it is possible that the different intracellular pools of beta-catenin are directed by the biological needs of the cell [39, 48]. For example, if beta-catenin is bound to APC in the cytoplasm, it will be targeted for degradation. However, if the formation of epithelial junctions is necessary, E-cadherin may recruit cytoplasmic beta-catenin to form adherens junctions. Further, the activation of cell signaling pathways that transmit their external signals to the nucleus would block the beta-catenin degradation machinery. This leads to the accumulation of cytosolic beta-catenin which then enters the nucleus where it binds with transcriptional co-activators to regulate gene transcription [39, 48]. In our studies, quercetin treated normal kidneys suggest that beta-catenin is recruited from the nucleus to the membrane to form adherens junctions. This was particularly evident in the ureteric epithelium in the 160μM treatment which was more intact compared to the 80μM treatment and demonstrated a much stronger beta-catenin membrane localization. In support of our data, other studies have shown that quercetin alters beta-catenin intracellular distribution. In cancer cells high nuclear beta-catenin levels were observed, whereas quercetin treated cells showed decreased nuclear beta-catenin and higher cytoplasmic and membrane expression [31]. However, future mechanistic studies will be required to determine how quercetin reduces nuclear beta-catenin and how quercetin facilitates beta-catenin translocation between the different intracellular pools.

A hallmark of renal dysplasia is reduced nephron number and is a risk factor for developing progressive renal disease and hypertension in children and adults [49, 50]. An 80% reduction in nephron number at birth results in a rapid development of proteinuria, hypertension, and renal failure [51, 52]. In rodents, the experimentally induced loss of nephron number during embryonic development or shortly after birth is strongly correlated to the progression of hypertension and kidney damage [53–55]. The renal histopathological phenotypes observed in $RD^B$ include regions of undifferentiated mesenchyme, stalled nephrogenic structures, and a significant reduction in the number of developing and maturing nephrons. These abnormalities are associated with improper expression levels and intracellular localization of the mesenchymal marker vimentin and epithelial markers NCAM and E-cadherin. The abnormal expression patterns of vimentin and E-cadherin are also observed in human dysplastic tissue. These results suggest that mesenchymal cells and early developing nephrons do not progress through proper mesenchymal-to-epithelial transition, which is essential for nephron formation [56]. In contrast, we demonstrated that treatment of dysplastic kidneys with quercetin resulted in a reduction in vimentin and enhanced NCAM and E-cadherin expression at the membrane, and an associated 44.5% increase in the number of properly developing and maturing nephrons. This is consistent with previous reports in which cancer cells treated with quercetin enhanced the transition of mesenchymal cells into epithelialized structures by downregulating the expression of mesenchymal markers and enhancing epithelial markers [57–59]. Further, epidermal cancer cells showed low levels and a diffuse cytoplasmic distribution of E-cadherin, whereas quercetin treated epidermal cancer cells demonstrated higher E-cadherin

levels and grew in clusters with more well-defined cellular junctions where E-cadherin expression is detected [57]. Therefore, our data supports that quercetin partially rescues nephrogenesis and increases nephron numbers by re-establishing the proper epithelialization of stalled nephrogenic structures and enhancing mesenchyme to epithelial transition in the undifferentiated mesenchyme cells.

The proper formation of cadherins junctions is essential for epithelial transition and nephron formation [60]. Our findings demonstrate that quercetin treatment of dysplastic kidneys results in beta-catenin and E-cadherin localizing to the cell membrane in transitioning mesenchymal cells and stalled nephrogenic structures. We suggest that this partial rescue in disrupted nephrogenesis is due to the proper formation of epithelial junctions. Similar to our $RD^B$ dysplastic kidneys, dysplastic tissue from oral squamous cancer showed mesenchyme-like malignant cells that have reduced beta-catenin and E-cadherin membrane expression [61]. The lack of beta-catenin and E-cadherin membrane localization resulted in a loss of attachment to the actin cytoskeleton, and the disassembly of these adherens junctions components led to a loss of epithelial integrity and polarity [62]. In further support of our findings, studies on human cancer cell lines have shown that quercetin increases the membranous localization of beta-catenin and E-cadherin, leading to improved cell adhesion and reduced metastasis [31, 63, 64]. These studies suggest that in our quercetin treated dysplastic kidneys, the re-establishment of beta-catenin and E-cadherin expression in membrane of the transitioning mesenchyme may also lead to a re-attachment to the actin cytoskeleton, therefore improving adherens junctions integrity and promoting epithelial transition [62]. Our findings also suggest that quercetin treated dysplastic kidneys could lead to increased beta-catenin/E-cadherin complex formation compared to untreated controls. It has not been shown in other disease models whether quercetin treatment leads to increased beta-catenin and E-cadherin binding. However, other plant-derived compounds that are structurally similar to quercetin, such as carnosol and tangeritin, have been demonstrated to functionally upregulate beta-catenin/E-cadherin complex formation [65]. Therefore, future experiments aimed at understanding the molecular mechanism of quercetin action in $RD^B$ dysplastic kidneys would confirm whether quercetin treatment increases beta-catenin and E-cadherin complex formation at the cell membrane.

Although renal dysplasia is the leading cause of childhood kidney disease, it remains a poorly understood disease and there is no cure. The *in vitro* studies presented here show that the abnormal levels of beta-catenin can be targeted and improve nephron formation in dysplastic kidneys. Our data and previous studies support that quercetin treatment reduces abnormal nuclear beta-catenin levels and re-distributes it to the cell membrane. This helps mediate cell adhesion in an E-cadherin dependent manner which improves junction integrity and facilitates epithelial transition in the stalled nephrons.

## Supporting information

**S1 Raw Images. Raw blot/gel images.** PDF file containing all raw, unedited and uncropped TIFF images for gel/blot results used for Fig 2. Three replicates of the Western blot experiment are shown (Replicate #1, #2 and #3). The Coomassie-stained gel used to demonstrate protein loading is also shown.
(PDF)

## Acknowledgments

We would like to thank Dr. Bradley Doble, Dr. Nathan Magarvey and Dr. Aftab Taiyab for their helpful discussions and suggestions; Saakethiya Sriranjan for assistance in

troubleshooting RT-PCR and Western blot protocols. Lily Morikawa for her expertise in embryonic kidney explant tissue processing and sectioning.

## Author Contributions

**Conceptualization:** Joanna Cunanan, Darren Bridgewater.

**Data curation:** Joanna Cunanan, Erin Deacon, Kristina Cunanan, Zifan Yang, Antje Ask, Lily Morikawa, Darren Bridgewater.

**Funding acquisition:** Darren Bridgewater.

**Investigation:** Darren Bridgewater.

**Methodology:** Joanna Cunanan, Lily Morikawa, Ekaterina Todorova, Darren Bridgewater.

**Project administration:** Antje Ask, Darren Bridgewater.

**Resources:** Darren Bridgewater.

**Supervision:** Antje Ask.

**Writing – original draft:** Joanna Cunanan, Darren Bridgewater.

**Writing – review & editing:** Joanna Cunanan, Darren Bridgewater.

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
