## [Decision Letter · Decision Letter 0]

7 Apr 2020

PONE-D-20-03892

Quercetin treatment reduces the severity of renal dysplasia in a beta-catenin dependent manner

PLOS ONE

Dear Dr Bridgewater,

Thank you for submitting your manuscript to PLOS ONE. After careful consideration, we feel that it has merit but does not fully meet PLOS ONE’s publication criteria as it currently stands. Therefore, we invite you to submit a revised version of the manuscript that addresses the points raised during the review process.

In particular, we would like you to provide additional biochemical evidence for the reported alterations in expression levels of vimentin and E-cadherin, and to clarify the statistical analyses. Furthermore, we would like you to expand your discussion to include additional previous related work, for instance by including some of the literature as suggested by reviewer 2. 

We would appreciate receiving your revised manuscript by May 22 2020 11:59PM. However, please notice, that we understand that at the moment you may not be able to perform experiments, and we are happy to extend this time for as long as you need. To enhance the reproducibility of your results, we recommend that if applicable you deposit your laboratory protocols in protocols.io, where a protocol can be assigned its own identifier (DOI) such that it can be cited independently in the future. For instructions see: http://journals.plos.org/plosone/s/submission-guidelines#loc-laboratory-protocols

We look forward to receiving your revised manuscript.

Kind regards,

Mirjam M Zegers, Ph.D.

Academic Editor

PLOS ONE

2. To comply with PLOS ONE submission requirements, in your Methods section, please provide additional information regarding the experiments involving animals and ensure you have included details on (1) the source of the mice, (2) the number of mice used and (3) methods of sacrifice of the dams.

In addition, please provide details regarding the source of the kidney tissue, including the date(s) during which kidney tissue was obtained and the source (nephrectomy, deceased donor, etc). Please also provide a table of relevant demographic details if available. Finally, in the ethics statement in the Methods and online submission information, please also ensure that you have specified (1) whether consent was informed and (2) what type was obtained (for instance, written or verbal, and if verbal, how it was documented and witnessed). If the need for consent was waived by the ethics committee, please include this information.

3. "All animal studies were performed in accordance with animal care guidelines (Animal Utilization Protocol #18-03-12)".

Please amend your current ethics statement to include the full name of the ethics committee that approved your specific study.

For additional information about PLOS ONE submissions requirements for ethics oversight of animal work, please refer to http://journals.plos.org/plosone/s/submission-guidelines#loc-animal-research  

5. Please include a caption for figure 4.

Reviewers' comments:

Reviewer's Responses to Questions

**Comments to the Author**

1. Is the manuscript technically sound, and do the data support the conclusions?

Reviewer #1: Yes

Reviewer #2: Yes

2. Has the statistical analysis been performed appropriately and rigorously? 

Reviewer #1: Yes

Reviewer #2: Yes

3. Have the authors made all data underlying the findings in their manuscript fully available?

Reviewer #1: Yes

Reviewer #2: Yes

4. Is the manuscript presented in an intelligible fashion and written in standard English?

Reviewer #1: Yes

Reviewer #2: Yes

5. Review Comments to the Author

Reviewer #1: The manuscript entitled “Quercetin treatment reduces the severity of renal dysplasia in a beta-catenin dependent manner” by Cunanan et al found that quercetin treatment reduced the nuclear localization of β- catenin in renal dysplasia. This study also identified that quercetin treatment increased the development of maturing neurons by improving the epithelial organization associated with higher E-cadherin levels. The manuscript provided very valuable observation and revealed the efficacy of quercetin in reducing renal dysplasia. However, some further studies are required to confirm the result provided, and need minor clarifications.

Major points

1. Figure 6 shows the expression levels of vimentin and E-cadherin in normal and dysplastic kidney tissue by immunostaining. Please include mRNA expression or protein expression levels of vimentin and E-cadherin in these samples for the conformation of immunostaining results.

2. Based on the data presented in the manuscript please discuss the effect of quercetin on β-catenin and E-cadherin complex formation, include relevant references also.

Minor points

1. Please indicate the method used for statistical analysis with a separate heading.

2. Here multiple samples were compared to control samples (Figure 2 G, H and I). The authors indicated that “data was analyzed using two-tailed Student’s t-test”. Usually, the t-test is used to compare two values and one-way Anova is useful to compare multiple values. Please confirm.

3. Please define the *, **, and *** in the figure legends (Figure 2)

Reviewer #2: In this work, the authors revealed that quercetin treatment attenuates the severity of renal dysplasia in a beta-catenin dependent manner. Several suggestions are made as follows to improve the quality of the manuscript.

1. Many references have been cited before 2010 especially references 29-39. β-catenin signaling pathway have been widely reported in the decade. The reviewer suggests the authors updated and cited the latest publications. I have searched the references for the different section of manuscript.

2. Please introduce more backgrounds about renal fibrosis treatment and explain why do you study quercetin? The bioactivities of quercetin have been widely reported many studies, such as Br J Pharmacol 2020,177(8):1841-1852; Redox Biol 2020,28:101337; Sci Rep 2019,9(1):19176; Free Radic Biol Med 2019,145:146-160; Free Radic Biol Med 2019,143:240-251; Phytomedicine 2019,57:65-71; Phytomedicine 2019,56:183-193; Sci Rep 2018,8(1):6194. Please summarize and cite them in the introduction section.

3. All abbreviations should be substantiated for the first time.

4. Natural products against renal injury via β-catenin signaling pathway have been widely reported by several recent reviews, such as Trends Pharmacol Sci 2018,39(11):937-952; Med Res Rev 2020,40(1):54-78; Biomed Pharmacother 2019,117:108990. Please summarize and cite them in the introduction section.

5. Chemicals, Reagents and antibodies should be provided. Please include both the manufacturer’s name and location (including city, state, and country) for specialized equipment and reagents throughout the manuscript.

6. β-catenin signaling pathway in Renal injury and fibrosis has been reported several articles such as Br J Pharmacol 2018,175(13):2689-2708; Phytomedicine 2018,42:207-218; Ther Adv Chronic Dis. 2019,10:2040622319869116; J Agric Food Chem 2018,66(8):1828-1842; Redox Biol 2017,12:505-521. Please discuss and cite them in the discussion section.

6. PLOS authors have the option to publish the peer review history of their article (what does this mean?). If published, this will include your full peer review and any attached files.

Reviewer #1: No

Reviewer #2: No

---

## [Author Response · Author response to Decision Letter 0]

6 May 2020

On behalf of my authors I wish to thank the editor and reviewers for their positive and supportive comments and for their thorough evaluation of our initial submission. We are very pleased to have addressed 100% of the editor’s and reviewers’ comments and suggestions, which has significantly enhanced our manuscript. Below, we restate the reviewer’s comments and respond to each in turn. 

Editor’s Comments: 

Comment 1: Please ensure that your manuscript meets PLOS ONE's style requirements, including those for file naming. The PLOS ONE style templates can be found at http://www.plosone.org/attachments/PLOSOne_formatting_sample_main_body.pdf and http://www.plosone.org/attachments/PLOSOne_formatting_sample_title_authors_affiliations.pdf

Response: The manuscript is formatted to the PLOS ONE submission guidelines. 

Comment 2: To comply with PLOS ONE submission requirements, in your Methods section, please provide additional information regarding the experiments involving animals and ensure you have included details on (1) the source of the mice, (2) the number of mice used and (3) methods of sacrifice of the dams. In addition, please provide details regarding the source of the kidney tissue, including the date(s) during which kidney tissue was obtained and the source (nephrectomy, deceased donor, etc). Please also provide a table of relevant demographic details if available. Finally, in the ethics statement in the Methods and online submission information, please also ensure that you have specified (1) whether consent was informed and (2) what type was obtained (for instance, written or verbal, and if verbal, how it was documented and witnessed). If the need for consent was waived by the ethics committee, please include this information.

Response: Our methods now include details on the mouse models and method of sacrifice (page 7 line 145-152). The ethics statement is now in full adherence to PLOS ONE guidelines. The human kidney tissue is from de-identified archived tissue from the Pathology Department at McMaster University. This use of this tissue is approved for research purposes in full adherence to guidelines from the Hamilton Health Sciences Integrated Research Ethics Board, approval #13-160-T. We have included amended ethics statements in our manuscript (Page 8 line 163-165) and submission forms. 

Comment 3: "All animal studies were performed in accordance with animal care guidelines (Animal Utilization Protocol #18-03-12)". Please amend your current ethics statement to include the full name of the ethics committee that approved your specific study. For additional information about PLOS ONE submissions requirements for ethics oversight of animal work, please refer to http://journals.plos.org/plosone/s/submission-guidelines#loc-animal-research. Once you have amended this/these statement(s) in the Methods section of the manuscript, please add the same text to the “Ethics Statement” field of the submission form (via “Edit Submission”).

Response: We included in the ethics statement that “The Hamilton Health Sciences Integrated Research Ethics Board” approved our animal studies (Animal Utilization Protocol #18-03-12) (Page 7 line 149-152). These amended statements are also included on the submission forms. 

Comment 4: PLOS ONE now requires that authors provide the original uncropped and unadjusted images underlying all blot or gel results reported in a submission’s figures or Supporting Information files. This policy and the journal’s other requirements for blot/gel reporting and figure preparation are described in detail at https://journals.plos.org/plosone/s/figures#loc-blot-and-gel-reporting-requirements and https://journals.plos.org/plosone/s/figures#loc-preparing-figures-from-image-files. When you submit your revised manuscript, please ensure that your figures adhere fully to these guidelines and provide the original underlying images for all blot or gel data reported in your submission. See the following link for instructions on providing the original image data: https://journals.plos.org/plosone/s/figures#loc-original-images-for-blots-and-gels. In your cover letter, please note whether your blot/gel image data are in Supporting Information or posted at a public data repository, provide the repository URL if relevant, and provide specific details as to which raw blot/gel images, if any, are not available. Email us at plosone@plos.org if you have any questions.

Response: We included all original uncropped and unadjusted images pertaining to Western blots in a supporting Information file. These images are generated directly from the GeneSys imaging system. To ensure accurate measurements of expression levels across all experimental trials, all blot images were taken using the same conditions (i.e. applied the same settings for exposure times and contrast levels during imaging to all three replicates). Coomassie-stained gel that represents total protein was used as the loading control. Specifically the 45 kDa band was cropped for Figure 2A. We have annotated each raw image that indicates the sample loading order and also demonstrates the bands excluded from analysis. The cover letter now indicates that the raw blot/gel images are in supporting files. 

Comment 5: Please include a caption for figure 4.

Response: We have added a complete caption for Figure 4. 

 

Reviewer #1 Comments: 

Comment 1: Figure 6 shows the expression levels of vimentin and E-cadherin in normal and dysplastic kidney tissue by immunostaining. Please include mRNA expression or protein expression levels of vimentin and E-cadherin in these samples for the conformation of immunostaining results.

Response: While it would be interesting to look at mRNA or protein levels, unfortunately we do not have access to fresh or frozen human kidney tissue. In previous studies we attempted in situ hybridization and RNA isolation from formalin fixed human kidney tissue obtained from pathology to look at GDNF expression in a previous study (Sarin, AJP, 2014). Despite numerous attempts we were not successful, due to unreliable RNA quality and yields. Based on this, and the limitation of de-identified archived formalin fixed paraffin embedded human kidney tissue, we feel we have planned and performed the most ideal experimental approach that best addresses our research question “determine the intracellular distribution of Vimentin and E-cadherin in human renal dysplasia”. The results from these IHC experiments nicely show Vimentin and E-cadherin intracellular expression at the protein level. 

Comment 2: Based on the data presented in the manuscript please discuss the effect of quercetin on β-catenin and E-cadherin complex formation, include relevant references also.

Response: We thank the reviewer for this thoughtful comment as this makes for an interesting discussion point, which we have included on page 29 line 670-677 of the revised manuscript. This is an interesting comment, since now that we have established that quercetin is effective in our model, our future studies will be aimed at understanding the molecular mechanisms of quercetin action in the kidney.

Comment 3: Please indicate the method used for statistical analysis with a separate heading.

Response: In the Methods Section we included a dedicated section to statistical analysis performed (page 12 line 265-273). 

Comment 4: Multiple samples were compared to control samples (Figure 2 G, H and I). The authors stated, “data was analyzed using two-tailed Student’s t-test”. Usually, the t-test is used to compare two values and one-way ANOVA is useful to compare multiple values. Please confirm.

Response: The reviewer is correct. A one-way ANOVA is a more appropriate test to analyze this experimental data. We reanalyzed the data in Fig 2G, H and I (and Fig1E, Fig 1N, Fig 2B). A Tukey’s post-hoc analysis determined which dose of quercetin are significantly different compared to the no quercetin group. We have updated this in the Methods and Results sections. 

Comment 5: Please define the *, **, and *** in the figure legends (Figure 2)

Response: We defined the asterisks in the Figure 2 captions and results section (and for all other figures with asterisks) to report the statistical significance.  

Reviewer #2 Comments: 

Comment 1: Many references have been cited before 2010 especially references 29-39. β-catenin signaling pathway have been widely reported in the decade. The reviewer suggests the authors updated and cited the latest publications. I have searched the references for the different sections of manuscript.

Response: We have included updated references on Wnt/beta-catenin signaling pathway in development and disease. However, we did maintain references from some of the earlier fundamental studies, especially those pertaining to the pathogenesis of renal dysplasia and beta-catenin/E-cadherin binding. 

Comment 2: Please introduce more background about renal fibrosis treatment and explain why do you study quercetin? The bioactivities of quercetin have been widely reported many studies, such as Br J Pharmacol 2020,177(8):1841-1852; Redox Biol 2020,28:101337; Sci Rep 2019,9(1):19176; Free Radic Biol Med 2019,145:146-160; Free Radic Biol Med 2019,143:240-251; Phytomedicine 2019,57:65-71; Phytomedicine 2019,56:183-193; Sci Rep 2018,8(1):6194. Please summarize and cite them in the introduction section.

Response: We agree that a more thorough background of quercetin actions and its role in renal fibrosis would add to the manuscript. Therefore, we added more information by Ren et al., 2016 (Sci Rep. 

2016;6:23968) to the introduction section (Page 5 Line 95-120). In addition, we included more detailed pharmacological bioactivities of quercetin and also included a clear rationale for focusing on quercetin in the kidney (Page 5 Line 95-120). In short, beta-catenin is overexpressed in the nucleus in dysplastic kidneys. This overexpression results in altered gene expression which contributes to the pathogenesis of renal dysplasia. Since previous studies in the kidney and other model systems demonstrated that quercetin was able to reduce beta-catenin and its transcriptional activity, we thought this would be a good candidate to start. Further, purified quercetin manufactured by Sigma Aldrich (#Q4951) is readily available and reasonably priced. It has been cited by many reputable studies that confirm its efficacy in decreasing nuclear beta-catenin and it has been utilized for use in tissue/organ cultures. 

Comment 3: All abbreviations should be substantiated for the first time.

Response: We have defined all abbreviations the first time they are used. 

Comment 4: Natural products against renal injury via β-catenin signaling pathway have been widely reported by several recent reviews, such as Trends Pharmacol Sci 2018,39(11):937-952; Med Res Rev 2020,40(1):54-78; Biomed Pharmacother 2019,117:108990. Please summarize and cite them in the introduction section.

Response: We agree that acknowledging other compounds that are used to reduce abnormal beta-catenin signaling in renal diseases should be included. Therefore, in the introduction we included a section on page 5 line 95-100 dealing with other plant derived natural compounds that are effective at reducing abnormal beta-catenin levels and signaling activity in renal fibrosis and chronic kidney disease. 

Comment 5: Chemicals, reagents and antibodies should be provided. Please include both the manufacturer’s name and location (including city, state, and country) for specialized equipment and reagents throughout the manuscript.

Response: All specialized materials, reagents, and equipment now includes the manufacturer name city, state, and country. 

Comment 6: β-catenin signaling pathway in Renal injury and fibrosis has been reported several articles such as Br J Pharmacol 2018,175(13):2689-2708; Phytomedicine 2018,42:207-218; Ther Adv Chronic Dis. 2019,10:2040622319869116; J Agric Food Chem 2018,66(8):1828-1842; Redox Biol 2017,12:505-521. Please discuss and cite them in the discussion section.

Response: Adding in these studies is a good idea to support our current study that demonstrate blocking abnormal beta-catenin signaling using quercetin prevents the expression of genes that contribute to pathogenesis and reduces the severity of kidney damage. These studies are included in the discussion on page 25 line 588-599.

---

## [Editor Report · Decision Letter 1]

27 May 2020

Quercetin treatment reduces the severity of renal dysplasia in a beta-catenin dependent manner

PONE-D-20-03892R1

Dear Dr. Bridgewater,

We are pleased to inform you that your manuscript has been judged scientifically suitable for publication and will be formally accepted for publication once it complies with all outstanding technical requirements.

With kind regards,

Mirjam M Zegers, Ph.D.

Academic Editor

PLOS ONE
---

## [Editor Report · Acceptance letter]

3 Jun 2020

PONE-D-20-03892R1 

Quercetin treatment reduces the severity of renal dysplasia in a beta-catenin dependent manner 

Dear Dr. Bridgewater:

I'm pleased to inform you that your manuscript has been deemed suitable for publication in PLOS ONE. Congratulations! Your manuscript is now with our production department. 

Kind regards, 

on behalf of

Dr. Mirjam M Zegers 

Academic Editor

PLOS ONE